# Improving Multimodal Sentiment Analysis: Supervised Angular Margin-based Contrastive Learning for Enhanced Fusion Representation

**Cong-Duy Nguyen**[1], **Thong Nguyen**[2], **Duc Anh Vu**[1], **Luu Anh Tuan** [1*]

[1]Nanyang Technological University, Singapore
[2]National University of Singapore, Singapore

nguyentr003@ntu.edu.sg, anhtuan.luu@ntu.edu.sg

## Abstract

The effectiveness of a model is heavily reliant on the quality of the fusion representation of multiple modalities in multimodal sentiment analysis. Moreover, each modality is extracted from raw input and integrated with the rest to construct a multimodal representation. Although previous methods have proposed multimodal representations and achieved promising results, most of them focus on forming positive and negative pairs, neglecting the variation in sentiment scores within the same class. Additionally, they fail to capture the significance of unimodal representations in the fusion vector. To address these limitations, we introduce a framework called Supervised Angular-based Contrastive Learning for Multimodal Sentiment Analysis. This framework aims to enhance discrimination and generalizability of the multimodal representation and overcome biases in the fusion vector's modality. Our experimental results, along with visualizations on two widely used datasets, demonstrate the effectiveness of our approach.

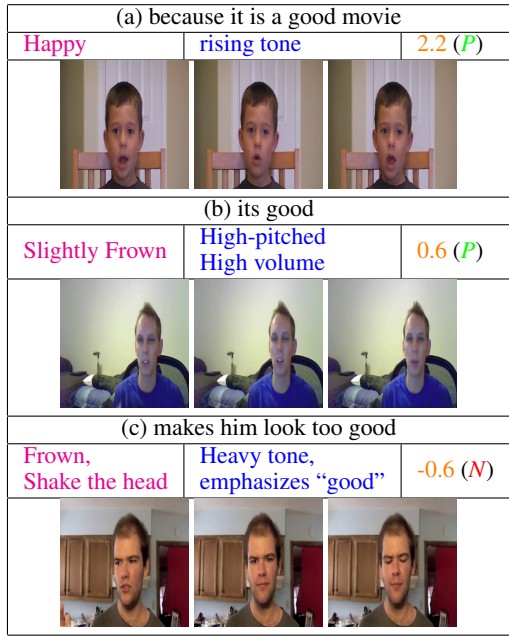

|  |  |  |
|---|---|---|
| (a) because it is a good movie | | |
| Happy | rising tone | 2.2 (*P*) |

|  |  |  |
|---|---|---|
| (b) its good | | |
| Slightly Frown | High-pitched High volume | 0.6 (*P*) |

|  |  |  |
|---|---|---|
| (c) makes him look too good | | |
| Frown, Shake the head | Heavy tone, emphasizes "good" | -0.6 (*N*) |

Table 1: Example in CMU-Mosi dataset. All three transcripts of the samples contain the word "Good," yet their sentiment scores span a range from highly positive at 3.0 to slightly negative at −0.6. (*P*) for positive class, (*N*) for negative class.

## 1 Introduction

Through internet video-sharing platforms, individuals engage in daily exchanges of thoughts, experiences, and reviews. As a result, there is a growing interest among scholars and businesses in studying subjectivity and sentiment within these opinion videos and recordings (Wei et al., 2023). Consequently, the field of human multimodal language analysis has emerged as a developing area of research in Natural Language Processing. Moreover, human communication inherently encompasses multiple modalities, creating a heterogeneous environment characterized by the synchronous coordination of language, expressions, and audio modalities. Multimodal learning leverages diverse sources of information, such as language (text/transcripts), audio/acoustic signals, and visual modalities (images/videos). This stands in contrast to traditional machine learning tasks that typically focus on single modalities, such as text or voice (Tay et al., 2017, 2018a,b).

As shown in Table 1, the primary objective of Multimodal Sentiment Analysis (MSA) is to utilize fusion techniques to combine data from multiple modalities in order to make predictions for the corresponding labels. In the context of emotion recognition and sentiment analysis, multimodal fusion is crucial since emotional cues are frequently distributed across different modalities. However, previous studies, as highlighted by (Hazarika et al., 2022), have pointed out that task-related information is not uniformly distributed among these modalities. Specifically, the text modality often exhibits a higher degree of importance and plays

---
*Corresponding Author

| | Video | Transcript | | |
|---|---|---|---|---|
| | | Face Expression | Voice | Sentiment Score |
| a |  | | I just got finished watching an excellent movie called "Mars needs moms" | |
| | | Extremely exciting | Enthusiastic Tone, High volume, emphasizes "excellent" | 3.0 (*P*) |
| b |  | | storyline was ok | |
| | | Unemotional | Normal volume Steady tone | 0.6 (*P*) |
| c |  | | oh god this is bad | |
| | | Angry, Stroking the face | Aggressive tone, High volume | -3.0 (*N*) |
| d |  | | and the only explanation i come up with makes this story kind of kind of creepy | |
| | | Frown | Monotone, Normal volume, Hesitant | -1.0 (*N*) |

Table 2: Example in CMU-Mosi dataset. The first two samples are positive, while the last two are negative. All four examples demonstrate diversity in sentiment representation. The first two examples belong to the positive group, but the first example exhibits happiness across all three modalities, whereas the second example only indicates positivity through the phrase "ok". This pattern is similarly observed in the negative example pair (c,d).

a more pivotal role compared to the visual and audio modalities. While words undoubtedly carry significant emotional information (e.g., words like "good", "great", "bad"), emotions are conveyed not only through words but also other communication channels, *e.g.* tone of voice or facial expressions.

We introduce a fusion scheme called Triplet loss for triplet modality to enhance representations of partial modalities in the fusion vector. In our study, we found that incorporating facial expressions and tone of voice is crucial in determining communication-related scores. Neglecting these modalities can lead to a loss of valuable information. For instance, in Table 1, we observe that even though all three transcripts convey positive sentiments about the film, the sentiment scores differ significantly. This difference arises from the variations in facial expressions and tone of voice. When a modality is hidden, it alters the representation and makes it challenging to perceive the underlying sentiment. By considering the interrelation among triplet modalities, we propose a self-supervised task to bridge the representation gap between the complete input and missing modalities. The objective is to ensure that an input with a concealed modality is more similar to the complete input than containing only one modality. This approach aims to improve the overall representation of multimodal data.

Previous research often focused on contrastive learning techniques, forming positive and negative representation pairs, but overlooked distinctions within the same class. As illustrated in Table 2,

despite belonging to the same class, the examples possess varying sentiment levels. Particularly, the first positive example (2a) has a score of 3.0, while the second positive example has a score of 0.6, indicating a significant difference. Similarly, the negative examples in (2c) and (2d) differ in sentiment. Therefore, these representations should not be considered closely related in the feature space. Furthermore, the distance between the (2a-c) pair should be greater than that of the (2a-d) pair, considering the sentiment.

To address this challenge, we challenge the assumption that all positive samples should be represented similarly in the feature space. In the process of sampling positive and negative pairs, we take into account the sentiment score difference between each sample. When the difference in sentiment score does not exceed a predefined threshold, we consider the input as positive. We propose a Supervised Angular Margin-based Contrastive Learning approach that enhances the discriminative representation by strengthening the margin within the angular space while capturing the varying degrees of sentiment within the samples. By not assuming uniform representation for all positive/negative samples in the feature space, our method effectively handles the differences between their sentiment levels.

Our contributions can be summarized as follows:

- We propose a novel approach called Supervised Angular Margin-based Contrastive

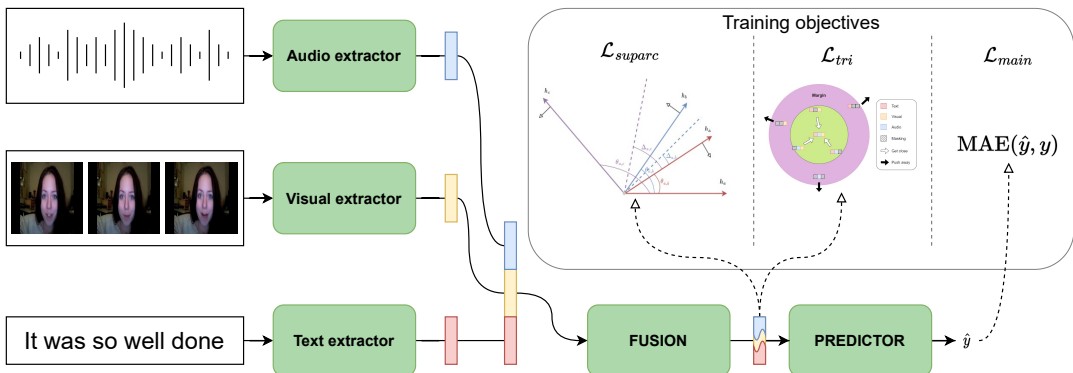

Figure 1: Implementation of our framework. The framework consists of three inputs for text, visual and audio modalities, finally, the whole model is train with three objectives: Supervised Angular Margin-based Contrastive Learning and Triplet Modalities Triplet lossl.

Learning for Multimodal Sentiment Analysis. This method enhances the discriminative representation of samples with varying degrees of sentiment, allowing for more accurate classification.

- We introduce a self-supervised triplet loss that captures the generalized representation of each modality. This helps to bridge the representation gap between complete inputs and missing-modality inputs, improving the overall multimodal fusion.

- Extensive experiments were conducted on two well-known Multimodal Sentiment Analysis (MSA) datasets, namely CMU-Mosi and CMU-Mosei. The empirical results and visualizations demonstrate that our proposed approach significantly outperforms the current state-of-the-art models in terms of sentiment analysis performance.

## 2 Related Work

Recently, multimodal research has garnered attention because of its many applications and swift expansion (Wang et al., 2019; Nguyen et al., 2022, 2023; Wei et al., 2022, 2024). In particular, multimodal sentiment analysis (MSA) predicts sentiment polarity across various data sources, like text, audio, and video. Early fusion involves concatenating the features from different modalities into a single feature vector (Rosas et al. 2013; Poria et al. 2016b). This unified feature vector is then employed as input. Late fusion entails constructing separate models for each modality and subsequently combining their outputs to obtain a final result (Cai and Xia 2015; Nojavanasghari et al. 2016).

Some works such as (Wöllmer et al., 2013) and (Poria et al., 2016a) have explored the utilization of hybrid fusion techniques that integrate both early and late fusion methods. In addition, MAG-BERT model (Rahman et al., 2020) combines BERT and XLNet with a Multimodal Adaptation Gate to incorporate multimodal nonverbal data during fine-tuning. Besides, (Tsai et al., 2020) introduced a novel Capsule Network-based method, enabling the dynamic adjustment of weights between modalities. MMIM (Han et al., 2021) preserves task-related information through hierarchical maximization of Mutual Information (MI) between pairs of unimodal inputs and the resulting fusion of multiple modalities. Recently, several works (Ma et al., 2021a; Sun et al., 2022; Zeng et al., 2022) have focused on uncertainly solving the missing modalities problem.

Contrastive Learning (Chopra et al., 2005) has led to its wide adoption across various problems in the field and is widely applied in many applications (Nguyen and Luu, 2021; Wu et al., 2022). Noteworthy training objectives such as N-Pair Loss (Sohn, 2016), Triplet Margin Loss (Ma et al., 2021b), Structured Loss (Song et al., 2016), and ArcCSE (Zhang et al., 2022), directly apply the principles of metric learning. In supervised downstream tasks, softmax-based objectives have shown promise by incorporating class centers and penalizing the distances between deep features and their respective centers. Center Loss (Wen et al., 2016), SphereFace (Liu et al., 2017), CosFace (Wang et al., 2018), and ArcFace (Deng et al., 2018) have become popular choices in deep learning applications in computer vision and natural language processing. However, these loss functions are designed specifically for classification tasks and are not appropriate

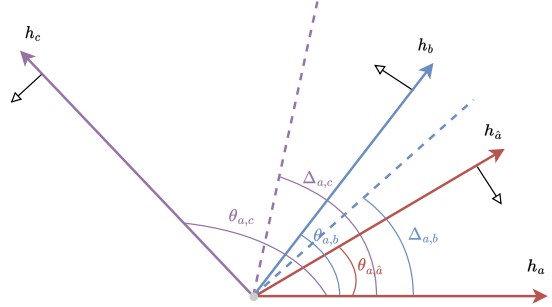

Figure 2: The framework of Supervised Angular Margin-based Contrastive Learning

for regression labels. Consequently, we put forward a novel training objective called SupArc, which exhibits enhanced discriminative capabilities when modeling continuous sentiment scores compared to conventional contrastive training objectives.

## 3 Methodology

### 3.1 Problem Statement

We aim to discern sentiments in videos by utilizing multimodal signals. Each video in the dataset is broken down into utterances—an utterance is a segment of speech separated by breaths or pauses. Every utterance, a mini-video in itself, is treated as an input for the model.

For each utterance U, the input is comprised of three sequences of low-level features from the language (t), visual (v), and acoustic (a) modalities. These are denoted as $U_t$, $U_v$, and $U_a$, each belonging to its own dimensional space, where $U_m \in \mathbb{R}^{l_m \times d_m}$, with $l_m$ represents the sequence length and $d_m$ represents the dimension of the modality $m$'s representation vector, $m \in t, v, a$. Given these sequences $U_m$ (where $m$ can be $t$, $v$, or $a$), the primary objective is to predict the emotional direction of the utterance $U$. This can be from a predefined set of $C$ categories, represented as $y$, or a continuous intensity variable, represented as $y$.

### 3.2 Model Architecture

As shown in Figure 1, each sample has three raw inputs: a text - transcript of video, video, and audio. Before feeding into the model, our model firstly processes raw input into numerical sequential vectors. We transform plain texts into a sequence of integers for the transcripts using a tokenizer. We use a feature extractor to pre-process raw format into numerical sequential vectors for visual and audio. Finally, we have the multimodal sequential

data $X_m \in \mathbb{R}^{l_m \times d_m}$.

### Mono-modal representation

$$h_t = \text{Text\_Encoding}\,(X_t) \qquad (1)$$
$$h_v = \text{Visual\_Encoding}\,(X_v) \qquad (2)$$
$$h_a = \text{Audio\_Encoding}\,(X_a) \qquad (3)$$

where $h_m \in \mathbb{R}^{d_m}$, $m \in t, v, a$, Visual and Audio _Encoding are bidirectional LSTMs and Text _Encoding is BERT model.

$$h = \text{FUSION}\,([h_t, h_v, h_a]) \qquad (4)$$

where FUSION module is a multi-layer perceptron and $[.,.,.]$ is the concatenation of three modality representations.

**Inference**  Finally, we take the output of fusion and feed it into a predictor to get a predicted score $\hat{y}$:

$$\hat{y} = \text{PREDICTOR}\,(h) \qquad (5)$$

where $\hat{y}$ ranges from $-3$ to $3$ and PREDICTOR module is a multi-layer perceptron.

## 4 Training objective

In this section, we introduce the objective function of our model with two different optimization objectives: **Supervised Angular Margin-based Contrastive Learning** and **Triplet Modalities Triplet loss**. The model's overall training is accomplished by minimizing this objective:

$$\mathcal{L} = \mathcal{L}_{\text{main}} + \alpha\,\mathcal{L}_{\text{suparc}} + \beta\,\mathcal{L}_{\text{tri}} \qquad (6)$$

Here, $\alpha, \beta$ is the interaction weights that determine the contribution of each regularization component to the overall loss $\mathcal{L}$. $\mathcal{L}_{\text{main}}$ is a main task loss, which is **mean absolute error**. Each of these component losses is responsible for achieving the desired subspace properties. In the next two sections, we introduce two different optimization objectives:

### 4.1 Supervised Angular Margin-based Contrastive Learning

This section explains the formulation and its derivation from our Supervised Angular Margin-based Contrastive Learning (SupArc).

In order to represent the pairwise associations between samples as positive or negative, we first

create fusion representations and categorize them into positive and negative pairs. These pairs are then used as input for a training objective we aim to optimize. In classical NLP problems - text-only tasks - this classification hinges on the specific relationships we wish to highlight - positive pairs may represent semantically similar or relevant sentences. In contrast, negative pairs typically consist of unrelated or contextually disparate sentences. In this multi-modal setting, we consider a fusion representation $h$ from equation 4. Previous works consider samples whose sentiment score is more significant than 0 (or less than 0) to be the same class in their objective function. However, we get each positive (and negative) sample in a batch by looking at the difference between the sentiment scores of the two examples. A pair of fusion representations considering to same class have its sentiment score difference under a threshold $TH$:

$$t_{i,j} = \begin{cases} 1 & \text{if } |y_i - y_j| \leq threshold \\ 0 & \text{other} \end{cases} \quad (7)$$

After gathering the positive and negative pairs, we put them into a training objective:

$$\mathcal{L} = -\log \frac{e^{sim(h_i, h_i^*)/\tau}}{\sum_{j=1}^{n} e^{sim(h_i, h_j)/\tau}} \quad (8)$$

where $sim$ is the cosine similarity, $\tau$ is a temperature hyper-parameter and $n$ is the number of samples within a batch, $i^*$ is a positive sample where $t_{i,i^*} = 1$.

Past research (Zhang et al., 2022) had proposed an ArcCos objective to enhance the pairwise discriminative ability and represent the entailment relation of triplet sentences:

$$\mathcal{L}_{arccos} = -\log \frac{e^{\phi(\theta_{i,i^*} + m)/\tau}}{e^{\phi(\theta_{i,i^*} + m)/\tau} + \sum_{j \neq i}^{n} e^{\phi(\theta_{j,i})/\tau}} \quad (9)$$

where angular $\theta_{i,j}$ is denoted as follow:

$$\theta_{i,j} = \arccos\left(\frac{h_i^T h_j}{\|h_i\| * \|h_j\|}\right) \quad (10)$$

and $m$ is an extra margin, $\phi$ is cos function $cos$.

However, there are differences between samples if considering examples 2c and 2c are both negative samples of example 2a, moreover in the fusion space, for example, the distance between, we modify the contrastive which can work on supervised

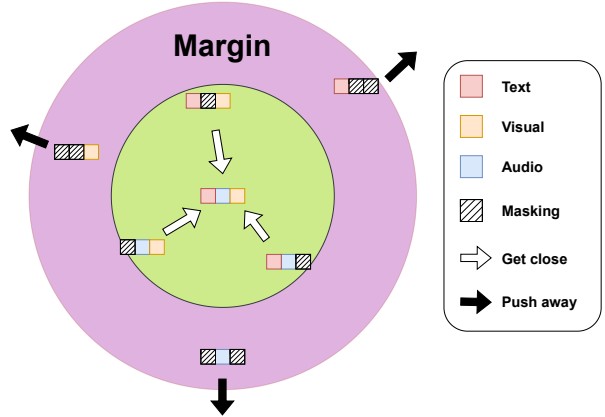

Figure 3: Implementation of Triplet Modalities Triplet loss.

dataset how can model the volume of the difference of a pair of samples. To overcome the problem, we propose a new training objective - Supervised Angular-based Contrastive Loss modified from 4.1 for fusion representation learning by adding a between opposing pair $h_i$ and $h_j$. Figure 2 illustrates our implementation, $\hat{a}$ is the positive sample of $a$, while $b, c$ are two negative samples. Moreover, the difference between pair $a - c$ is more significant than $a - b$. Thus, the margin (dashed line) of $c$ (purple) is further than $b$ (blue). The objective function called **SupArc** is formulated as follows:

$$\mathcal{L}_{suparc} = -\log \frac{e^{\phi(\theta_{i,i^*})/\tau}}{e^{\phi(\theta_{i,i^*})/\tau} + \sum_{j \neq i}^{n} e^{\phi(\theta_{i,j} - m\Delta_{i,j})/\tau}} \quad (11)$$

where $\Delta_{i,j} = |y_i - y_j|$ is the difference of sentiment score between two samples $i$ and $j$.

## 4.2 Triplet Modalities Triplet loss

Previously the training objectives only considered the fusion representation between different samples with similarities or differences in features within a given space. In reality, each type of input (visual, text, audio) plays a vital role in predicting the sentiment score. For example, examples 1a, 1b, 1c all have almost similar compliments (containing "good" in the sentence), thus if we consider only text in this task, the sentiment scores should not be too much different, however, with the difference between facial expressions (visual) and phonetic-prosodic properties (audio), the sentiment scores now are varying degrees. Thus, losing one or two modalities before integration can cause lacking information. Existing methods cannot handle such

problems between modalities and can cause a bias problem in one modality (Hazarika et al., 2022).

To distinguish the slight differences in semantics between different sentences, we propose a new self-supervised task that models the entailment relation between modalities of an input. For each triplet modality $\{t, v, a\}$ in the dataset, we will mask one and two in three modalities before feeding into the fusion module. We will have $h_{\backslash x}$ for masking one modality $x$ and $h_{\backslash x,y}$ for masking two $x, y$ of three.

$$h_{\backslash x} = \text{FUSION}\left([0 * h_x, h_y, h_z]\right) \qquad (12)$$

$$h_{\backslash x,y} = \text{FUSION}\left([0 * h_x, 0 * h_y, h_z]\right) \qquad (13)$$

As masking only one modality will keep more information than masking two, thus, in the fusion space, it must be nearer to the complete. $h$ is more similar to $h_{\backslash x}$ than $h_{\backslash x,y}$. As illustrated in Figure 3, three vectors that are losing one modality (one masking) must be closed to the completed one in the center, while others with only one modality left have to be moved far away. We apply a triplet loss as an objective function:

$$L_{tri} = \sum_{x,y \in (t,v,a)} max(0, s(h, h_{\backslash x,y}) \qquad (14)$$
$$- s(h, h_{\backslash x}) + m_{tri})$$

where $s(a, b)$ is the similarity function between $a$ and $b$ - we use cosine similarity in this work, $m_{tri}$ is a margin value.

# 5 Experimental Setup

In this part, we outline certain experimental aspects such as the datasets, metrics and feature extraction.

## 5.1 Datasets

We perform experiments on two open-access datasets commonly used in MSA research: CMU-MOSI (Zadeh et al., 2016) and CMU-MOSEI (Bagher Zadeh et al., 2018).

**CMU-MOSI** This dataset comprises YouTube monologues where speakers share their views on various topics, primarily movies. It has a total of 93 videos, covering 89 unique speakers, and includes 2,198 distinct utterance-video segments. These utterances are hand-annotated with a continuous opinion score, ranging from -3 to 3, where -3 signifies strong negative sentiment, and +3 indicates strong positive emotion.

**CMU-MOSEI** The CMU-MOSEI dataset expands upon the MOSI dataset by including a greater quantity of utterances, a wider range of samples, speakers, and topics. This dataset encompasses 23,453 annotated video segments (utterances), derived from 5,000 videos, 1,000 unique speakers, and 250 diverse topics.

## 5.2 Metrics

In our study, we adhere to the same set of evaluation metrics that have been persistently used and juxtaposed in past research: Mean Absolute Error (MAE), Pearson Correlation (Corr), seven-class classification accuracy (Acc-7), binary classification accuracy (Acc-2), and F1 score.

The Mean Absolute Error (MAE) is a popularly used metric for quantifying predictive performance. It calculates the average of the absolute differences between the predicted and actual values, thereby providing an estimate of the magnitude of prediction errors without considering their direction. It gives an intuition of how much, on average, our predictions deviate from the actual truth values.

Pearson Correlation (Corr) is a statistical measure that evaluates the degree of relationship or association between two variables. In our context, it indicates the extent of skewness in our predictions. This measure is particularly crucial as it helps understand if our prediction model overestimates or underestimates the true values.

The seven-class classification accuracy (Acc-7) is another significant metric. It denotes the ratio of correctly predicted instances that fall within the same range of seven defined intervals between -3 and +3, in comparison with the corresponding true values. This metric is essential for understanding how accurately our model can classify into the correct sentiment intervals.

Furthermore, we also employ binary classification accuracy (Acc-2), which provides insights into how well our model performs in predicting whether a sentiment is positive or negative. It is computed as the ratio of the number of correct predictions to the total number of predictions.

Lastly, the F1 score is computed for positive/negative and non-negative/negative classification results. This metric is a harmonic mean of precision and recall, providing a balanced measure of a model's performance in a binary or multi-class classification setting. It's particularly useful when the data classes are imbalanced, helping us assess

| Models | CMU-MOSI | | | | | CMU-MOSEI | | | | |
|--------|------|-----|------|-------|-------|------|-----|------|-------|-------|
|        | MAE | F1 | Corr | Acc-7 | Acc-2 | MAE | F1 | Corr | Acc-7 | Acc-2 |
| ICCN | 0.862 | - /83.0 | 0.714 | 39.0 | - /83.0 | 0.565 | - /84.2 | 0.713 | 51.6 | - /84.2 |
| MulT | 0.861 | 80.6/83.9 | 0.711 | - | 81.5/84.1 | 0.580 | - /82.3 | 0.703 | - | - /82.5 |
| MISA | 0.804 | 80.77/82.03 | 0.764 | - | 80.79/82.10 | 0.568 | 82.67/83.97 | 0.724 | - | 82.59/84.23 |
| MAG-BERT | 0.731 | 82.6/84.3 | 0.789 | - | 82.5/84.3 | 0.539 | 83.7/85.1 | 0.753 | - | 83.8/85.2 |
| Self-MM | 0.713 | 84.42/85.95 | 0.798 | - | 84.00/85.98 | 0.530 | 82.53/85.30 | 0.765 | - | 82.81/85.17 |
| MIMM | 0.700 | 84.00/85.98 | 0.800 | 46.65 | 84.14/86.06 | 0.526 | 82.66/**85.94** | 0.772 | 54.24 | 82.24/**85.97** |
| Ours | **0.679** | **84.32/86.25** | **0.806** | **48.11** | **84.40/86.28** | **0.520** | **84.71**/85.82 | **0.774** | **55.01** | **84.57**/85.97 |

Table 3: Results on CMU-MOSI and CMU-MOSEI are as follows. We present two sets of evaluation results for Acc-2 and F1: non-negative/negative (non-neg) (left) and positive/negative (pos) (right). The best results are indicated by being marked in bold. Our experiments, which are underlined, have experienced paired t-tests with $p < 0.05$ and demonstrated overall significant improvement.

| Models | MAE | F1 | Corr | Acc-7 | Acc-2 |
|--------|-----|-----|------|-------|-------|
| Our model | 0.520 | 84.71/85.82 | 0.774 | 55.01 | 84.57/85.97 |
| w/o $\mathcal{L}_{suparc}$ | 0.527 | 79.82/83.83 | 0.766 | 54.61 | 79.21/83.76 |
| w/o $\mathcal{L}_{tri}$ | 0.523 | 83.53/85.41 | 0.771 | 54.37 | 83.24/85.63 |
| w/o $\mathcal{L}_{suparc}, \mathcal{L}_{tri}$ | 0.532 | 81.87/85.28 | 0.758 | 54.21 | 81.40/85.32 |

Table 4: Ablation results on CMU-MOSEI.

the effectiveness of our model in handling such scenarios.

## 5.3 Feature Extraction

For fair comparisons, we utilize the standard low-level features that are provided by the respective benchmarks and utilized by the SOTA methods.

### 5.3.1 Language Features

Previous studies used GloVe embeddings for each utterance token, however, recent works have started to apply a pre-trained transformers model (like BERT) on input transcripts. We also follow this setting which uses a BERT tokenizer - a WordPiece tokenizer. It works by splitting words either into full forms (e.g., one word becomes one token) or into word pieces — where one word can be broken into multiple tokens.

### 5.3.2 Visual Features

The MOSI and MOSEI datasets use **Facet** to gather facial expression elements, including facial action units and pose based on FACS. Facial expressions are vital for understanding emotions and sentiments, as they serve as a primary means of conveying one's current mental state. Smiles are particularly reliable in multimodal sentiment analysis, and OpenFace, an open-source tool, can extract and interpret these visual features. This process is performed for each frame in the video sequence. The dimensions of the visual features are $d_v$, with 47 for MOSI and 35 for MOSEI.

### 5.3.3 Acoustic Features

Multimodal sentiment analysis relies on crucial audio properties like MFCC, spectral centroid, spectral flux, beat histogram, beat sum, most pronounced beat, pause duration, and pitch. Extracted using COVAREP, these low-level statistical audio functions contribute to the 74-dimensional acoustic features ($d_a$) in both MOSI and MOSEI. These features comprise 12 Mel-frequency cepstral coefficients, pitch, VUV segmenting attributes, glottal source parameters, and more related to emotions and speech tonality.

## 5.4 Settings

We use BERT (Devlin et al., 2018) (BERT-base-uncased) as the language model for the text extractor, we load the BERT weight pre-trained on Book-Corpus and Wikipedia from Pytorch framework Huggingface, and we also use BERT tokenizer for feature extractor on text, on top of the BERT, we apply an MLP to project the hidden representation into smaller space. For acoustic and visual, we use 1-layer BiLSTM with feature sizes of input are 74, 36, respectively, and the hidden state's sizes are both 32. The fusion module is the MLP, with an input size of 32x3 (text, visual, acoustic) and for fusion vector size is 32. We set $\alpha = 0.1, \beta = 0.1$ for both objectives. We trained our model is trained with a learning rate $l_r = 1e^{-4}$ in 12 epochs using AdamW as an optimizer; we set the batch size of 32 on 1 V100 GPU and trained about 2-4 hours for CMU-MOSEI and about 30 minutes for CMU-MOSI.

## 6 Experimental Results

### 6.1 Baseline

We conducted a comparison between our method and other baselines

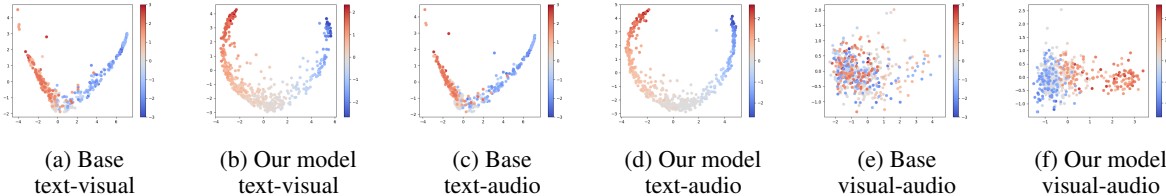

|  (a) Base
text-visual | (b) Our model
text-visual | (c) Base
text-audio | (d) Our model
text-audio | (e) Base
visual-audio | (f) Our model
visual-audio |

Figure 4: Visualization of the masking fusion representation in the testing set of the MOSEI dataset using PCA technique. We conduct on 3 pairs: text-visual, text-audio, visual-audio on 2 models: base model and ours.

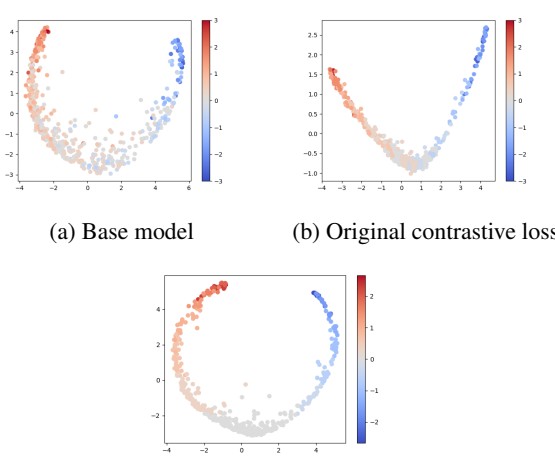

(a) Base model      (b) Original contrastive loss

(c) Our model

Figure 5: Visualization of the fusion representation in the testing set of the MOSEI dataset using PCA technique. We conduct on 3 three settings: (a) base model, (b) original contrastive loss, (c) our model.

- Multimodal Transformer **MulT** (Tsai et al., 2019) develops an architecture that incorporates separate transformer networks for unimodal and crossmodal data. The fusion process is achieved through attention mechanisms, facilitating a comprehensive integration of the different modalities.

- Interaction Canonical Correlation Network **ICCN** (Sun et al., 2019) minimizes canonical loss between modality representation pairs to ameliorate fusion outcome.

- Modality-Invariant and -Specific Representations **MISA** (Hazarika et al., 2020) involves projecting features into two distinct spaces with specific constraints, one for modality-invariant representations and the other for modality-specific representations.

- Multimodal Adaptation Gate for BERT **MAG-BERT** (Rahman et al., 2020) designs and integrates alignment gate into the BERT model to enhance and fine-tune the fusion process.

- Self-supervised Multi-Task Learning **SELF-MM** (Yu et al., 2021) assigns a specific uni-modal training task to each modality, utilizing automatically generated labels. This work aims to modify gradient back-propagation.

- Multimodal-informax **MMIM** (Han et al., 2021): utilizes a two-level mutual information (MI) maximization approach to synthesize fusion results from multi-modality.

## 6.2 Results

Following the methodology of previous studies, we conducted five times of our model using the same hyper-parameter settings. The average performance is presented in Table 2. Our findings indicate that our model achieves better or comparable results compared to various baseline methods. To provide further details, our model demonstrates a significant performance advantage over the state-of-the-art (SOTA) in all metrics on CMU-MOSI, as well as in MAE, Corr, Acc-7, Acc-2 (non-neg), and F1 (non-neg) scores on CMU-MOSEI. Regarding others, our model exhibits very similar performance on Acc-2 (pos) and is slightly lower than MMIM on F1 (pos). These results offer initial evidence of the effectiveness of our approach in addressing this task.

## 7 Analysis

### 7.1 Regularization

To validate the significance of our objectives, we select top-performing models for each dataset and gradually remove one loss at a time. Setting the corresponding variables ($\alpha$, $\beta$) to 0 nullifies a specific loss. Results are presented in Table 4, indicating that incorporating all losses leads to the best performance. While the triplet loss shows a slight improvement, the contrastive loss significantly enhances performance. These findings suggest that distinct representations of fusion spaces are truly beneficial. Combining the triplet and contrastive

losses can further improve performance. The triplet loss helps avoid modality bias (e.g., text), thus optimizing fusion space when used in conjunction with contrastive loss.

## 7.2 Visualizing Contrastive Representation

When applying contrastive objectives, it is important to analyze the generalization of these characteristics. To do so, we visualize fusion vector representations for test samples. Figure 5 shows that the angular-based contrastive loss performs better than the base model and original contrastive. Both contrastive losses effectively map vectors for sentiment similarity, unlike the base model's representations, which lack clarity.

## 7.3 Visualizing Masked-modality fusion

We visualize masked modality fusion vectors to compare different representations when removing a modality. Following previous settings, we visualize fusion vectors with one modality masked out for two models: the base model and our approach. In Figures 4, the text-visual and text-audio patterns resemble the complete version. However, the base model's representation has a heavy overlap between data points, while our method shows more generalization. For the audio-visual pair, both models poorly represent the features, but the base model mixes data points with different label scores. On the contrary, our model with dual training objectives separates neutral points and divides positive/negative nuances into two distinct groups.

## 8  Conclusion

In conclusion, our framework, Supervised Angular-based Contrastive Learning for Multimodal Sentiment Analysis, addresses the limitations of existing methods by enhancing discrimination and generalization in the multimodal representation while mitigating biases in the fusion vector. Through extensive experiments and visualizations, we have demonstrated the effectiveness of our approach in capturing the complexity of sentiment analysis. The publicly available implementation of our framework paves the way for further research and development in this field, benefiting applications such as opinion mining and social media analysis.

## 9  Limitations

Our model works on two popular datasets: CMU-Mosi and CMU-Mosei. These datasets are cat-

egorized as multimodal sentiment analysis task, which is inherited from traditional text-based sentiment analysis tasks. However, we are only working on annotation with sentiment in the range $[-3, 3]$ while traditional text-based sentiment analysis can extend as an emotion recognition which is a multi-class classification task. We have not studied our methods on emotion and we acknowledge these areas as opportunities for future research, aiming to enhance our framework's optimization in various contexts and utilize it in a wider range of applications.

## Acknowledgement

We thank all anonymous reviewers for their helpful comments. This research is supported by the National Research Foundation, Singapore under AI Singapore Programme, AISG Award No: AISG2-TC-2022-005 and by the Singapore Ministry of Education (MOE) Academic Research Fund (AcRF) Tier 1 (RS21/20).

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
