# OpenReview forum: "Improving Multimodal Sentiment Analysis: Supervised Angular margin-based Contrastive Learning for Enhanced Fusion Representation"
_EMNLP/2023/Conference — EMNLP 2023 Findings_

### Official Review · Reviewer_kyoU · 2023-08-03

**Soundness:** 4

**Excitement:**

4: Strong: This paper deepens the understanding of some phenomenon or lowers the barriers to an existing research direction.

**Missing References:**

Can add more references related to this work.

**Paper Topic And Main Contributions:**

The paper is about Supervised Angular-based Contrastive Learning for Multimodal Sentiment Analysis to improve discrimination and generalization of multimodal representation. The contributions of the paper include: proposed supervised Angular Margin-based Contrastive Learning for Multimodal Sentiment Analysis, introduced self-supervised triplet loss that captures the generalized representation of each modality, and extensive experimental setup.

**Questions For The Authors:**

A. Explain the analysis clearly especially from figure 4 and figure 5.

**Reasons To Accept:**

A. Proposed angular margin-based contrastive learning.
B. Fusion of Self-supervised triplet loss is good.
B. The paper is written well.
c. EXperimental setup and analyis are good.

**Reasons To Reject:**

A. Related work can be improved.
B. The insights are not clearly explained from the analysis

**Reproducibility:**

4: Could mostly reproduce the results, but there may be some variation because of sample variance or minor variations in their interpretation of the protocol or method.

**Reviewer Confidence:**

4: Quite sure. I tried to check the important points carefully. It's unlikely, though conceivable, that I missed something that should affect my ratings.

---

> ### Author Rebuttal · Authors · 2023-08-28
>
> Thank you for your helpful comments!
>
> We're glad that you appreciate our well-written paper and comprehensive experiments.
>
> We sincerely hope our response can address your concerns and improve your ratings.
>
> ===================================
>
> **_Q1: Concerns about the Related work_**
>
> Thank you for your suggestion! We'll add more important references in the final revision.
>
> ===================================
>
> **_Q2: Figure 4 explanation_**
>
> We extract the fusion features and visualize these vectors for test samples using PCA. **_Base_** means the model with only task loss. The text-visual and text-audio representations resemble the complete version (like in Figure 5). However, when we mask out text modality - the most important modality, although, both models poorly represent the features, the base model shows useless representations. Our model can create two discrete positive/negative groups.
>
> ===================================
>
> **_Q3: Figure 5 explanation_**
>
> With the same visualization setting from experiments in Figure 4, Figure 5 shows that our supervised angular-based contrastive loss performs better than the base model. We can see that all 3 models show generalization. However, in the base model, there are samples with different sentiment points that overlap quite a lot, for example, the slightly red and gray parts or the bluish and gray parts. In original contrastive learning, the feature representation is better when the overlap is less, the examples form a continuous and regular color range. Lastly, our method shows better feature representations, and better generalizability, and puts forward each different sample pair when adding the difference between two label scores to the contrastive loss.

---

### Official Review · Reviewer_yH9j · 2023-08-04

**Soundness:** 2

**Excitement:**

3: Ambivalent: It has merits (e.g., it reports state-of-the-art results, the idea is nice), but there are key weaknesses (e.g., it describes incremental work), and it can significantly benefit from another round of revision. However, I won't object to accepting it if my co-reviewers champion it.

**Paper Topic And Main Contributions:**

The main topic of the paper is in the area of fusing representations from multiple modalities to have better representations. The contributions are:
- A novel approach called “Supervised Angular Margin Based Contrastive learning for multimodal sentiment analysis” which increases the discriminative representation of samples with different degrees of sentiments. This is introduced as a loss function where there are two additional regularizers.
- The paper introduces a “self supervised triplet loss” in order to generalize representations of each modality (text, audio, video)


**Questions For The Authors:**

- What is the main motivation of the triplet loss? Is it mostly for cases when there is a missing modality you can still accurately predict the correct sentiment? Or to avoid over reliance/spurious correlations with one modality? Or both?


**Reasons To Accept:**

- Considering the varying degree of difference between sentiment scores compared to just sampling points from the positive and negative classes
- preventing over reliance on a given modality by using the triplet loss


**Reasons To Reject:**

-

**Reproducibility:**

3: Could reproduce the results with some difficulty. The settings of parameters are underspecified or subjectively determined; the training/evaluation data are not widely available.

**Reviewer Confidence:**

1: Not my area, or paper was hard for me to understand. My evaluation is just an educated guess.

**Typos Grammar Style And Presentation Improvements:**

- line 312 seems like there is a word missing =  "to overcome the problem, we propose a new training objective - supervised angular-based contrastive loss modified from 4.1 for fusion representation learning by adding a <missing_term> between opposing par hi and hj"?

---

> ### Author Rebuttal · Authors · 2023-08-28
>
> Thank you for your helpful comments!
>
> We're glad that you appreciate our well-written paper and comprehensive experiments.
>
> We sincerely hope our response can address your concerns and improve your ratings.
>
> ===========================================================================================
>
> **_Q1: What is the main motivation of the triplet loss? Is it mostly for cases when there is a missing modality you can still accurately predict the correct sentiment? Or to avoid over reliance/spurious correlations with one modality? Or both?_**
>
> Thank you for your question!
>
> Our triplet loss, designed for triplet modalities, endeavors to enhance the representations of partial modalities within the fusion vector. This enhancement is crucial for preventing the model from exhibiting bias towards a single modality. Prior research [1] has demonstrated that numerous Multimodal Sentiment Analysis (MSA) models often exhibit a bias towards the text modality.
>
> Our visualization in Figure 4 effectively demonstrates the efficacy of our proposed method. Upon masking out the text modality, both the base model and our model exhibit poor feature representations. However, a notable disparity arises in the fusion space: the base model's representations appear to be ineffectual, whereas our model successfully generates two distinct positive and negative clusters.
>
> This visualization underscores the superiority of our approach in mitigating modality bias and improving the overall quality of the learned representations.
>
> ===========================================================================================
>
> **_Q2: Grammar Style And Presentation Improvements_**
>
> We thank you for pointing them out, and we will definitely correct them in the revision.
>
> ===========================================================================================
>
> **References:**
>
> [1]. Analyzing Modality Robustness in Multimodal Sentiment Analysis NAACL 2022

---

### Official Review · Reviewer_qMbV · 2023-08-11

**Typos Grammar Style And Presentation Improvements:** 1. Figure 3 has not been referenced i…
**Soundness:** 3

**Excitement:**

2: Mediocre: This paper makes marginal contributions (vs non-contemporaneous work), so I would rather not see it in the conference.

**Missing References:**

1. https://arxiv.org/abs/1707.07250
2. https://aclanthology.org/W19-4331/
3. Many works published from 2021-present

**Paper Topic And Main Contributions:**

The authors aim to capture the significance of unimodal representations in the fusion vector using sentiment scores.  This framework aims to enhance discrimination and generalizability of the multimodal representation and overcome biases in the fusion vector’s modality using Margin-based Contrastive Learning.

**Questions For The Authors:**

How do you compare your results on the effect of various modalities with some old results on the significance of each single modality in emotion detection tasks? you can refer to some results on the above-mentioned works.

**Reasons To Accept:**

1. Choosing an important problem to focus on and solve
2. nice extension of an existing method to multimodal representation learning and fusion.


**Reasons To Reject:**

1. missing some methods including some old methods: https://arxiv.org/abs/1707.07250 , and https://aclanthology.org/W19-4331/ where study effect of dimension and factorization on prediction tasks with applications on the datasets studied in this draft.

2. related work section is very weak, and not covering SOTA. you have to add various integration methods focusing to solve multimodal integration issues.

3. The main idea already exists, and this work is a simple extension of the previously proposed method, with some simple intuitions.

**Reproducibility:**

3: Could reproduce the results with some difficulty. The settings of parameters are underspecified or subjectively determined; the training/evaluation data are not widely available.

**Reviewer Confidence:**

3: Pretty sure, but there's a chance I missed something. Although I have a good feel for this area in general, I did not carefully check the paper's details, e.g., the math, experimental design, or novelty.

---

> ### Author Rebuttal · Authors · 2023-08-28
>
> Thank you for your helpful comments!
>
> We're glad that you appreciate our well-written paper and comprehensive experiments.
>
> We sincerely hope our response can address your concerns and improve your ratings.
>
> ===========================================================================================
>
> **_Q1: missing some methods including some old methods_**
>
> Thank you for your suggestion. We will add those pioneer studies in the related work section of the revision.
>
> ===========================================================================================
>
> **_Q2: Related work section_**
>
> Thank you for your suggestion! We'll add the important references and others to the related work section. In 2022, UniMSE is proposed and get SOTA on CMU-MOSI, CMU-MOSEI, and also other two emotion recognition tasks. We achieved SOTA on these two tasks; moreover, their SOTA on CMU-MOSI and CMU-MOSEI reported in the UniMSE paper are MMIM () too. However, UniMSE trained their model with large datasets - almost double the amount of training data compared to our method before evaluation:
>
> |      |           |                 |  CMU-MOSI |           |                 |           |                 | CMU-MOSEI |           |                 |
> |------|:---------:|:---------------:|:---------:|:---------:|:---------------:|:---------:|:---------------:|:---------:|:---------:|:---------------:|
> |      | MAE       | F1              | Corr      | Acc7      | Acc2            | MAE       | F1              | Corr      | Acc7      | Acc2            |
> | Uni* | 0.693     | 83.98/85.91     | 0.802     | 47.56     | 84.04/85.96     | 0.523     | 84.56/85.55     | 0.771     | 54.29     | 83.24/85.27     |
> | Ours | **0.679** | **84.32/86.25** | **0.806** | **48.11** | **84.40/86.28** | **0.520** | **84.71/85.82** | **0.774** | **55.01** | **84.57/85.97** |
>
> (*) the results are obtained based on the official published code at https://github.com/LeMei/UniMSE.
>
> In general, our method achieves better performance compared to UniMSE[1]. We will include this comparison in the revision.
>
> ===========================================================================================
>
> **_Q3: The main idea already exists._**
>
> We proposed a novel contrastive objective, and it’s different from previous approaches. ArcFace [2] focuses loss function used in face recognition tasks, and ArcCse [3] is for sentence representation learning. Both two losses have fixed m which is a margin. Our supervised contrastive objective flexibly adapts m with the difference between anchor and negative samples. Furthermore, we are the first who apply angular-based contrastive learning to regressive tasks, from there, we can extend this technique to different problems not just MSA.
>
> ===========================================================================================
>
> **_Q4: Figure 3 has not been referenced in the text._**
>
> Thank you for pointing them out, we will add the Figure 3 reference in section 4.2 in the final revision.
>
> ===========================================================================================
>
> **References:**
>
> [1]. UniMSE: Towards Unified Multimodal Sentiment Analysis and Emotion Recognition EMNLP(2022)
>
> [2]. ArcFace: Additive Angular Margin Loss for Deep Face Recognition
>
> [3]. A Contrastive Framework for Learning Sentence Representations from Pairwise and Triple-wise Perspective in Angular Space

---

### Official Review · Reviewer_qpZv · 2023-08-12

**Soundness:** 2

**Excitement:**

3: Ambivalent: It has merits (e.g., it reports state-of-the-art results, the idea is nice), but there are key weaknesses (e.g., it describes incremental work), and it can significantly benefit from another round of revision. However, I won't object to accepting it if my co-reviewers champion it.

**Missing References:**

Missing Modality Imagination Network for Emotion Recognition with Uncertain Missing Modalities.
Tag-assisted Multimodal Sentiment Analysis under Uncertain Missing Modalities.
Counterfactual Reasoning for Out-of-distribution Multimodal Sentiment Analysis.

**Paper Topic And Main Contributions:**

This paper focuses on the Multimodal Sentiment Analysis task and proposes a framework called Supervised Angular-based Contrastive Learning for Multimodal Sentiment Analysis. This framework aims to enhance discrimination and generalizability of the multimodal representation and overcome biases in the fusion vector’s modality.

**Questions For The Authors:**

See weakness.

**Reasons To Accept:**

1. A novel framework is proposed to enhance the discriminative representation of samples.
2. The authors introduce a self-supervised triplet loss that captures the generalized representation of each modality.
3. This paper is well-organized.

**Reasons To Reject:**

1. The authors say that ''Although previous methods have proposed multimodal representations and achieved promising results, most of them focus on forming positive and negative pairs, neglecting the variation in sentiment scores within the same class'', do you mean previous MSA research mainly use contrastive learning for representation learning?
2. I can't find equation 3.2 because of the format mismatch. Do you mean equation (4)?
3. The benefit of the loss function defined in equation (11) should be detailed. Why it can model the volume of the difference of a pair of samples?
4. The modality-losing problem has been solved by several previous works, but the authors don't discuss them. For example, Tag-assisted Multimodal Sentiment Analysis under Uncertain Missing Modalities and Missing Modality Imagination Network for Emotion Recognition with Uncertain Missing Modalities.
5. More recent baselines should be compared, such as Counterfactual Reasoning for Out-of-distribution Multimodal Sentiment Analysis.
6. The performance improvement in the experimental section is not significant.

**Reproducibility:**

4: Could mostly reproduce the results, but there may be some variation because of sample variance or minor variations in their interpretation of the protocol or method.

**Reviewer Confidence:**

5: Positive that my evaluation is correct. I read the paper very carefully and I am very familiar with related work.

---

> ### Author Rebuttal · Authors · 2023-08-28
>
> Thank you for your helpful comments!
> We're glad that you appreciate our well-written paper and comprehensive experiments.
> We sincerely hope our response can address your concerns and improve your ratings.
>
> ===========================================================================================
>
> **_Q1: Do you mean previous MSA research mainly uses contrastive learning for representation learning?_**
>
> Thank you for your question!
> What we mean is that previous works only focus on the difference between positive class and negative class, but not on how different they are within a class. For example, in MMIM, they build two normal distributions for positive (score > 0) and negative (score < 0)  samples to calculate the entropy of a multivariate normal distribution but do not consider the differences between samples in the same class.
>
> ===========================================================================================
>
> **_Q2: I can't find equation 3.2 because of the format mismatch. Do you mean equation (4)?_**
>
> Yes, it should be equation (4). Thank you for pointing them out, and we will correct them in the revision.
>
> ===========================================================================================
>
> **_Q3: The benefit of the loss function defined in equation (11) should be detailed. Why it can model the volume of the difference of a pair of samples?_**
>
> In ArcFace and ArcCSE, the authors want to push away the negative sample with m degree using m (extra margin for decision boundary in angular-based CL). However, m is fixed for all anchor-negative pairs in the batch. Our SupArc uses the volume between label scores to scale m to adaptively push away negative samples with different extra margins. For instance, if we get a pair of examples 2a and 2b (Table 2), we will have an extra margin of 2.4m (3.0 - 0.6) while taking a pair of 2a and 2c, the extra margin is now 6.0m (3.0 - -3.0).
>
> ===========================================================================================
>
> **_Q4: The modality-losing problem has been solved by several previous works, but the authors don't discuss them._**
>
> Thank you for your suggestion! We will discuss them in the related work section of the revision.
>
> ===========================================================================================
>
> **_Q5: More recent baselines should be compared, such as Counterfactual Reasoning for Out-of-distribution Multimodal Sentiment Analysis_**
>
> Thank you for your suggestion. Among three related papers [1,2,3] that you suggested, only two papers [1,2] address the same tasks (CMU-MOSI, CMU-MOSEI) while the others solve the problem of Multimodal Emotion Recognition.
> As in [1], the authors only reported two metrics (i.e., Acc and F1) on 7-class classification and 2-class classification, while [2] reported only the accuracy of 2-class classification. We provide the comparison as follows:
>
> |      |  |           | CMU-MOSI      |           |           |  |           | CMU-MOSEI       |           |           |
> |------|:--------:|:---------:|:-----:|:---------:|:---------:|:---------:|:---------:|:-----:|:---------:|:---------:|
> |      | MAE      | F1        | Corr  | Acc7      | Acc2      | MAE       | F1        | Corr  | Acc7      | Acc2      |
> | TATE [2] | -        | -         | -     | -         | 84.90     | -         | -         | -     | -         | -         |
> | CLUE [1] | -        | 84.38     | -     | 48.04     | 84.31     | -         | 84.46     | -     | 53.42     | 84.52     |
> | Ours | 0.679    | **86.25** | 0.806 | **48.11** | **86.28** | 0.520     | **85.82** | 0.774 | **55.01** | **85.97**
>
>
> In general, our method achieves better performance compared to [1,2]. We will include this comparison in the revision.
>
> ===========================================================================================
>
> **_Q6: The performance improvement in the experimental section is not significant_**
>
> We conducted the significant test t-tests with $p<0.05$ (caption of Table 3). It proves that our proposed method achieves significant improvement compared to the baselines.
>
> ===========================================================================================
>
> **References:**
>
> [1]. Counterfactual Reasoning for Out-of-distribution Multimodal Sentiment Analysis
>
> [2]. Tag-assisted Multimodal Sentiment Analysis under Uncertain Missing Modalities
>
> [3]. Missing Modality Imagination Network for Emotion Recognition with Uncertain Missing Modalities

---

### Meta-Review · Area_Chair_GGws · 2023-09-21

**Recommendation:** 3

**Metareview:**

The paper introduces a supervised angular-based contrastive learning for multimodal sentiment analysis and shows how the proposed framework can attain more discriminate and generalized representation from each modality and overcome any bias towards a single modality by enhancing partial modality representation after fusion. The authors show the efficacy of the proposed framework with widely used sentiment datasets: CMU-Mosi and CMU-Mosei and compare with some state-of-the-art model performances.

The challenges addressed in the paper are well-researched, and the proposed framework brings together the ideas and techniques to apply them in MSA.

The reviews received for the paper are mixed. The majority of the reviewers point out the following concerns:
– The paper has a very weak related section, missing recent and old approaches proposed to solve these challenges.
– The presented idea is a simple extension and lacks sufficient novelty.
– Some insights are not well-explained in the paper.


The authors addressed some of the concerns in the rebuttal and added a new baseline. The author committed to improve the related work section in future versions of the paper.

Overall the framework proposed for SA, is a simple solution that shows comparable performance compared to  UniMSE (2022) and MIMM (2021).  The framework and the results in the paper can of be of interest to the researchers. We recommend that the authors take the suggestions from the reviews to further strengthen their work.

---

### Decision · Program_Chairs · 2023-10-07

**Decision:**

Accept-Findings

**Comment:**

The paper introduces a supervised angular-based contrastive learning for multimodal sentiment analysis and shows how the proposed framework can attain more discriminate and generalized representation from each modality and overcome any bias towards a single modality by enhancing partial modality representation after fusion. The authors show the efficacy of the proposed framework with widely used sentiment datasets: CMU-Mosi and CMU-Mosei and compare with some state-of-the-art model performances.

The challenges addressed in the paper are well-researched, and the proposed framework brings together the ideas and techniques to apply them in MSA.

The reviews received for the paper are mixed. The majority of the reviewers point out the following concerns:
– The paper has a very weak related section, missing recent and old approaches proposed to solve these challenges.
– The presented idea is a simple extension and lacks sufficient novelty.
– Some insights are not well-explained in the paper.


The authors addressed some of the concerns in the rebuttal and added a new baseline. The author committed to improve the related work section in future versions of the paper.

Overall the framework proposed for SA, is a simple solution that shows comparable performance compared to  UniMSE (2022) and MIMM (2021).  The framework and the results in the paper can of be of interest to the researchers. We recommend that the authors take the suggestions from the reviews to further strengthen their work.